# Defining the Underlying-Event Activity in the Presence of Heavy-Flavour Processes in Proton-Proton Collisions at LHC Energies

**László Gyulai [1,2]**, **Szende Sándor [1,3,*]** and **Róbert Vértesi [1]**

1    Wigner Research Centre for Physics, P.O. Box 49, H-1525 Budapest, Hungary; gyulai.laszlo@wigner.hu (L.G.); vertesi.robert@wigner.hu (R.V.)
2    Faculty of Natural Sciences, Budapest University of Technology and Economics, Műegyetem rkp. 3., H-1111 Budapest, Hungary
3    Faculty of Sciences, Eötvös Loránd University, Pázmány Péter Sétány 1/A, H-1117 Budapest, Hungary
*    Correspondence: sandor.szende@wigner.hu

**Abstract:** We present a systematic analysis of heavy-flavour production in the underlying event in connection to a leading hard process in pp collisions at $\sqrt{s} = 13$ TeV, using the PYTHIA 8 Monte Carlo event generator. We compare results from events selected by triggering on the leading hadron, as well as those triggered with reconstructed jets. We show that the kinematics of heavy-flavour fragmentation complicates the characterisation of the underlying event, and the usual method which uses the leading charged final-state hadron as a trigger may wash away the connection between the leading process and the heavy-flavour particle created in association with that. Events triggered with light or heavy-flavour jets, however, retain this connection and bring more direct information on the underlying heavy-flavour production process, but may also import unwanted sensitivity to gluon radiation. The methods outlined in the current work provide means to verify model calculations for light and heavy-flavour production in the jet and the underlying event in great details.

**Keywords:** ultra-relativistic collisions; LHC; heavy flavour; jets; underlying event



## 1. Introduction

It has long been anticipated that at sufficiently high temperatures and energy densities, matter converts into a phase where the quarks are not confined into hadrons [1], the so-called quark-gluon plasma (QGP). Contrary to the original expectations according to which the QGP would behave as an ideal gas, experiments at RHIC found a strong collective behaviour in the final state of heavy-ion collisions [2] that exhibits a scaling property with the number of constituent quarks [3]. This can be interpreted as hydrodynamical behaviour of a strongly coupled, fluid-like QGP where the degrees of freedom are the quarks [4]. Surprisingly, LHC measurements also observed collectivity in small colliding systems such as proton–proton (pp) and proton–lead (p–Pb) collisions with high final-state multiplicity [5,6]. Although in this case the creation of QGP in a small volume cannot be ruled out, several theoretical works explain collective behaviour in small systems with vacuum-QCD processes at the soft boundary, e.g., multiple-parton interactions (MPI) [7,8] or rope hadronisation with string shoving [9].

A traditional assumption that is used in collisions of small hadronic systems with a hard process is that the final state can be factorised into the results of the hard parton-parton scattering (jets) and the underlying event (UE)—the rest of the particles, which include secondary, softer processes as well as beam remnants. According to this picture, the UE is represented by the particle production in regions that fall further away from the direction determined by the QCD process with the highest transverse-momentum exchange. The particle production in this so-called transverse region is largely independent of this

leading process (usually a back-to-back jet pair). The relative transverse event-activity classifier $R_T$ [10] is often used to categorise events based on the underlying-event activity. In models that rely on MPI to simulate particle production, a strong correlation is observed between $R_T$ and the number of MPI processes in an event [11].

The observed collectivity in small systems suggests that soft or semi-soft processes are not negligible and may have an influence on jet development. Although a deconfined medium cannot exist in a substantial volume and therefore direct attenuation of the jets is not expected, modifications in the UE in connection to the leading jet may be a signature of such effects [12,13]. Measurements of unidentified hadron yields as a function of $R_T$ showed that the number of particles produced in the UE is almost independent of the momentum scale of the leading hard process [14,15]. However, identified probes with a production mechanism linked to the hard process may show such dependence and therefore be a tool to test the connection between the leading process and the underlying event.

Heavy quarks (charm and beauty) are mostly produced in the initial hard processes of high-energy hadron collisions. Their mean lifetime is longer than the duration of the QGP phase, so using these penetrating probes in small systems may help better understand the whole evolution of the system [16]. Proton–proton collisions serve as a baseline for heavy-ion measurements, therefore heavy-flavour production in these systems has to be understood in detail. There are colour-charge as well as mass-dependent differences between heavy and light-flavour production and fragmentation. While the majority of light-flavour jets are initiated by hard gluons, heavy-flavour jets mostly originate from quarks that are produced in the initial hard process. On the other hand, quark-mass dependent effects can be evaluated, both in the initial production and in the parton shower influenced by the dead-cone effect [17,18]. Measuring and modelling the creation of heavy flavour at the semi-soft boundary can help with differentiating between colour-charge and mass effects. While heavy-flavour leading processes have been modelled in terms of the UE [19], a systematic study of identified heavy-flavour production in association with a hard process (in a similar fashion to that in [14] for light flavour) has not yet been done.

In this work, we use PYTHIA 8 [20] simulations with jet reconstruction algorithms to show that $R_T$-differential measurements are an excellent experimental tool to study the connection between the underlying event and the initial hard process in a highly differential way. We argue that the traditional definition of $R_T$, based on a charged-hadron trigger, may not be ideal for heavy-flavour studies, and we propose using the hardest jet as a trigger to represent the leading process. We compare heavy-flavour production in the jet and the UE regions in charged-hadron triggered, light-jet triggered, as well as identified heavy-flavour jet triggered events. We argue that the current study, besides providing predictions with the particular model that we use, also gives physics insight and methodological recommendations that are valid beyond the considered model, and can motivate new experiments. Once high-precision experimental data are available, they can be compared with other models that focus on the collective aspects of hadronization, such as EPOS 3 [21] and or rope hadronisation with string shoving [9].

## 2. Analysis Method

We simulated pp collisions at $\sqrt{s} = 13$ TeV centre-of-mass energy using the Monte Carlo event generator PYTHIA 8 [20] version 8.235 with the Monash tune [22] and the NNPDF2.3 LO PDF set [23], with soft QCD settings and the MPI-based colour-reconnection scheme. Settings not listed here were kept at their default values. Motivated by the capabilities of the ALICE experiment [24], we considered charged final-state hadrons from PYTHIA in the pseudorapidity window $|\eta| < 0.8$ and above the transverse-momentum threshold $p_T = 0.15$ GeV/$c$. Particles with a lifetime $c\tau > 10$ mm can be typically tracked in the detector and they were therefore considered final. We studied hadron-triggered as well as jet-triggered events. In the first case, at least one final-state charged hadron ($\pi^\pm$, K$^\pm$, p or $\bar{\text{p}}$) was required in central pseudorapidity window $|\eta| < 0.8$ to have a transverse momentum above the threshold $p_T^{\text{trig}} = 5$ GeV/$c$. (The terms leading hadron and trigger

hadron are often used interchangeably for the final-state hadron with the highest $p_T$). In the case of jet triggers, we reconstruct charged-particle jets from the PYTHIA tracks with the anti-$k_T$ algorithm [25] using the FastJet package [26] with a resolution parameter $R = 0.4$. (Note that $R$ influences the amount of background picked up by the jet clustering algorithm as well as the fraction of the full parton shower is typically contained in the jet. This analysis uses $R = 0.4$, a choice employed by several inclusive and heavy-flavour jet analyses [27,28]). The jets were required to be fully contained within the pseudorapidity region of $|\eta| < 0.8$. The transverse-momentum threshold for the leading jet is $p_T^{\text{ch.jet,trig}} = 10\,\text{GeV}/c$. In the case of the jet trigger, we also identify charm and beauty jets (where the respective heavy quark is present within the jet cone) as well as light jets (where no heavy-flavour quark is present in the cone, indicating that the jet was initiated by either a gluon or an up, down, or strange quark).

Following the definition by the CDF Collaboration [10], the direction of the trigger hadron or jet is used to define regions in the azimuthal plane that have different sensitivity to the underlying event. We distinguish three spatial regions using the azimuthal angle: $|\Delta\varphi| < \frac{\pi}{3}$ for the toward region, $|\Delta\varphi| > \frac{2\pi}{3}$ for the away region, and $\frac{\pi}{3} < |\Delta\varphi| < \frac{2\pi}{3}$ for the transverse region. The UE activity can then be expressed through the charged-particle multiplicity in the transverse region via the transverse event activity classifier $R_T = \frac{N_{trans}}{\langle N_{trans}\rangle}$, where the transverse multiplicity $N_{trans}$ is the number of charged hadrons in the transverse region, and $\langle N_{trans}\rangle = 7.426$ in the current simulations. We classify events based on $R_T$ values into four categories: 0–0.5, 0.5–1, 1–2, 2–10, and we also use the $R_T$-integrated category (0–10) for reference.

We computed the yields of the heavy-flavoured non-strange prompt D mesons ($D^0$, $\overline{D}^0$, $D^\pm$ and $D^{*\pm}$, commonly referred to as D mesons in this text) and B mesons ($B^0$, $\overline{B}^0$, $B^\pm$ and $B^{*\pm}$) in terms of the underlying-event activity. Only primary heavy-flavour mesons were selected based on their PDG code [29]. Note that in the experiment, most of these channels are routinely reconstructed through their hadronic decays [30]. We also evaluated the yields of c and b quarks on the partonic level, to assess the effect of fragmentation. An experimental equivalent of this would be to measure the yield of heavy-flavour jets in association with a high-momentum trigger.

We simulated a total of 1 billion proton–proton events at $\sqrt{s} = 13$ TeV. The number of events with hadron trigger was 10.756 million, and those with jet triggers was 8.887 million, showing that the hadron-trigger and the jet-trigger conditions select hard processes on a similar momentum scale. Figure 1 (left) shows the simulated $R_T$ distributions for events triggered with the leading hadrons as well as those triggered with inclusive jets.

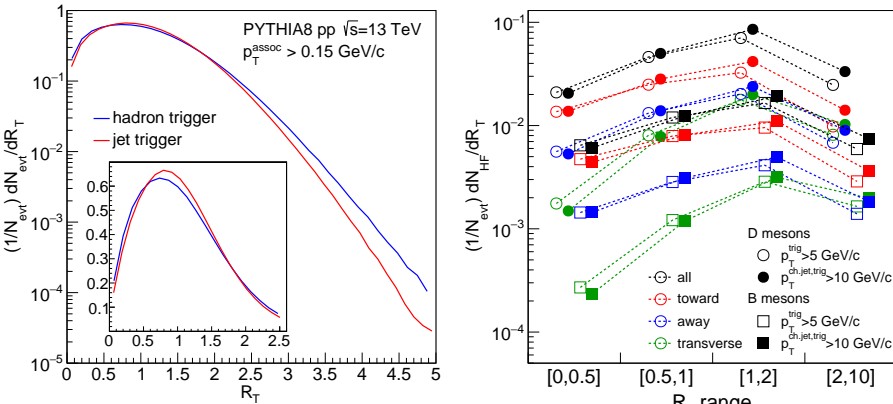

**Figure 1. Left**: $R_T$ probability distributions for events triggered with the leading hadrons ($p_T^{\text{trig}} > 5\,\text{GeV}/c$) compared to those triggered with inclusive jets ($p_T^{\text{ch.jet,trig}} > 10\,\text{GeV}/c$). The insert shows the same distribution on a linear scale. **Right**: Distributions of heavy-flavour D and B hadrons among the different $R_T$ classes in hadron-triggered as well as in jet-triggered events, shown for the full event as well as for the toward, transverse and away regions separately. (The dashed lines are to guide the eye).

As seen before [14], PYTHIA 8 reproduces the bulk of the $R_T$ distribution of hadron-triggered events, while it underestimates the high-$R_T$ tail. One can observe that the $R_T$ distribution corresponding to jets is somewhat more compact than the distribution of hadron-triggered events. This can be attributed to the fact that inclusive-jet triggers tend to select the leading process more reliably than charged-hadron triggers. Figure 1 (right) shows the production of D as well as B mesons corresponding to each $R_T$ category in case of hadron and of jet triggers, for the full event as well as for the toward, transverse and away regions. As expected, heavy-flavour production is more prominent in events with high underlying-event activity, especially in the transverse region.

## 3. Results

### 3.1. Charged Particle Production with Hadron and Jet Triggers

Experimental results from ALICE and CMS [14,15] show that both the number of charged hadrons and the sum of the transverse momenta in the toward and away regions increases with $p_T^{trig}$, as a consequence of jet fragmentation. The number of charged particles produced in the transverse region is, however, virtually independent of the trigger-hadron momentum and forms a plateau above $p_T^{trig} \approx 5\,\text{GeV}/c$. The sum of the transverse momenta carried by charged hadrons in this region shows a weak linear dependence on $p_T^{trig}$ above the same threshold, possibly because of a hard gluon radiation into the underlying event [31]. This behaviour is reproduced well by PYTHIA 8. A small discrepancy of 5–10% is present around the trigger threshold, although this can be partially taken care of with tuning [15]. Figure 2 (left) shows the average charged-particle multiplicity as a function of transverse momentum of the trigger hadron, from our simulations, compared to data from ALICE [14]. This behaviour is qualitatively repeated if one uses jet triggers instead of hadron triggers, as shown in Figure 2 (right). One notable difference is that the plateau in the transverse region is present only above $p_T^{ch.jet,trig} \approx 10\,\text{GeV}/c$, which is a consequence of the hadron trigger carrying only part of the transverse momentum of the fragmented jet.

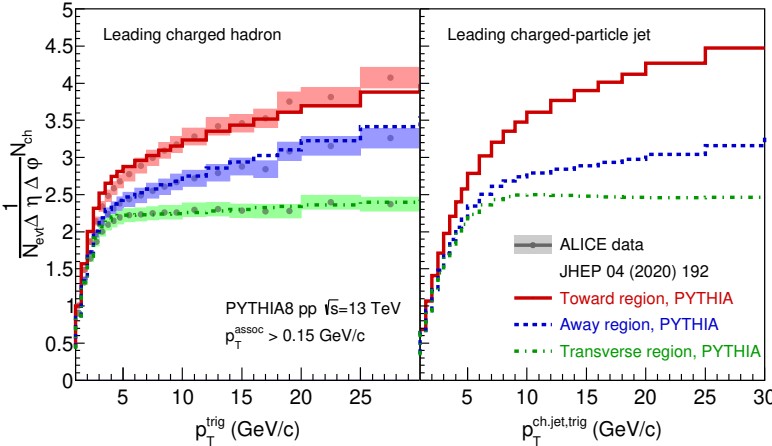

**Figure 2.** Charged-particle multiplicity in the toward, away and transverse regions as a function of the leading charged hadron (**left**) and charged-particle jet (**right**).

### 3.2. Heavy-Flavour Production with Hadron Triggers

In analyses that focus on light flavour production, the underlying event is often categorised using the highest-$p_T$ charged hadron as the trigger [14,15]. This convenient choice relies on the assumptions that the relevant fragmentation functions are similar in these events, and that in hard jets a charged hadron will typically carry a substantially large portion of the jet momentum. Hard enough jets will therefore be well represented by the leading charged hadron, and the sample will not be biased by the choice of the trigger. These assumptions work well for light-flavour-dominated samples. The situation, however, can be radically different if a high-$p_T$ heavy quark is involved. Since the gluon-radiation

phase space is limited by the dead-cone [17], heavy-flavour fragmentation is harder than that of light flavour: most of the momentum will end up in the heavy-flavour hadron, that in turn decays further into charged hadrons. Whether one of these hadrons will provide a trigger or not will depend on the fragmentation of that specific parton flavour as well as the kinematics of the relevant decay channels. This unfortunately means that $R_T$-dependent measurements of heavy-flavour particles will be specific to the given heavy-flavour hadron, and the results will not compare directly to light-flavour results. To overcome this problem, we define $R_T$ and the corresponding toward, away and transverse regions within the event using the axis of the leading jet. Although this definition should yield comparable results regardless of the type of the initiating parton, this definition is not without problems either, as we will see later.

For consistency with the experimental results [14,15] we utilised a hadron trigger threshold $p_T^{\text{trig}} = 5\,\text{GeV}/c$. Figure 3 shows the production of D mesons in the toward (top left) and transverse (top right) regions in events with hadron triggers above the threshold. The number of D mesons is normalised with the number of triggered events in each $R_T$ interval. In the toward region the per-trigger production of high-momentum D mesons ($p_T \gtrsim 8\,\text{GeV}/c$) shows no dependence on the transverse-event activity beyond uncertainties. These high-$p_T$ D mesons are mostly produced in the hard scattering and are not influenced by the underlying event. (A slight increase towards high transverse momentum in the highest $R_T$ bin is likely caused by the bias towards higher transverse-event activity when multiple $c\bar{c}$ pairs are produced). For D mesons with $p_T < 5\,\text{GeV}/c$, below the trigger threshold, a distinct dependence on the $R_T$ is observed. In this $p_T$-range, where D mesons are solely produced in softer processes, the per-trigger yield of D mesons increases with $R_T$, while its dependence on the $p_T$ is weak. From the trigger threshold upwards, the trigger gradually "turns on", reflecting the decay kinematics of the D mesons: while only about 16% of D mesons within $5 < p_T < 6\,\text{GeV}/c$ provide a $p_T^{\text{trig}} > 5\,\text{GeV}/c$ charged-hadron trigger, this value rises to 66% in the $7 < p_T < 8\,\text{GeV}/c$ range, and up to above 97% for D mesons with $p_T > 10\,\text{GeV}/c$. In the transverse region, a clear dependence of per-trigger production on $R_T$ is seen in the full $p_T$ range.

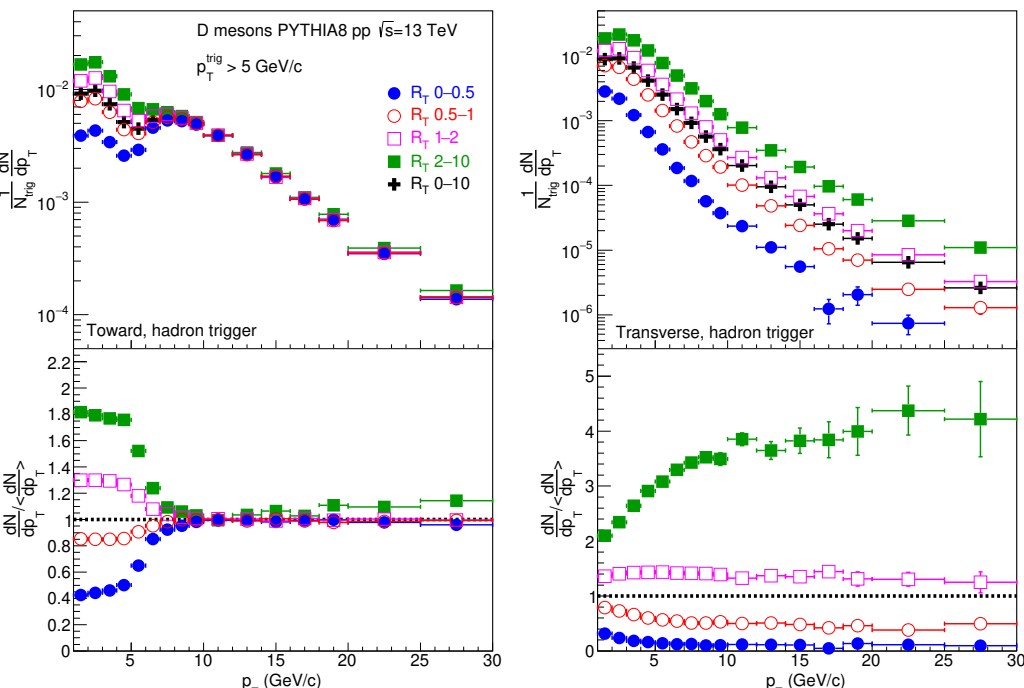

**Figure 3.** Per-trigger D-meson production with charged-hadron triggers as a function of $p_T$ for different $R_T$ classes in the toward (**left**) and transverse (**right**) regions. The **top** panels show the yields, while the **bottom** panels show their ratios over the $R_T$-integrated per-trigger yield.

The ratios of the $R_\mathrm{T}$-dependent production to the $R_\mathrm{T}$-integrated one are shown in Figure 3 in the bottom left and bottom right for the toward and transverse regions respectively. In the toward region, the D mesons are connected with the triggered hard process with an increasing frequency toward high $p_\mathrm{T}$, therefore the separation in $R_\mathrm{T}$ ranges vanishes. In the transverse region, however, the separation increases towards higher $p_\mathrm{T}$. This trend is qualitatively similar to that observed for unidentified charged hadrons in preliminary ALICE results [32].

### 3.3. Heavy-Flavour Production with Jet Triggers

Using a hadron trigger, one cannot distinguish between heavy and light-flavour jets that fragment into a high-$p_\mathrm{T}$ charged hadron, which makes it difficult to establish the direct connection between heavy-flavour production and the underlying event. A way around this is to separately look at events that are triggered with identified heavy-quark and light-flavour jets. In the following we evaluate D-meson production with charm (c) and light-flavour (u,d,s,g) triggers. In the c-jet triggered case we trigger on a hard charm jet and we are looking for a D meson. This may be part of the c-jet that balances the trigger, or may also come from a separately produced $c\bar{c}$ pair. On the other hand, with triggering on a light flavour we can exclude that the D meson comes from the same hard quark pair as the trigger: it will either come from gluon splitting, or from an entirely different, but softer process. To highlight the relative trends, in further figures we only show the ratios of D meson per-trigger yields in each $R_\mathrm{T}$ interval over the $R_\mathrm{T}$-integrated yield.

The left-hand side panels in Figure 4 show per-trigger D-meson production with c-jet triggers in the toward (top left) and transverse (bottom left) regions.

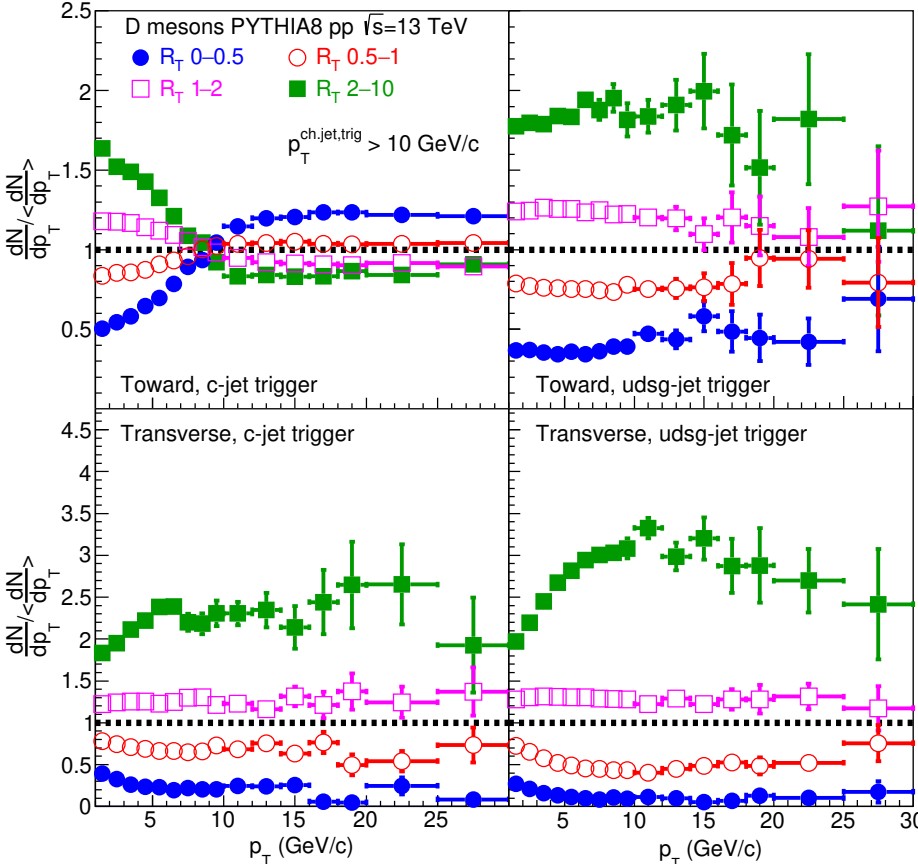

**Figure 4.** Per-trigger D-meson production ratios with charm-jet triggers (**left** panels) as well as light-jet triggers (**right** panels), as a function of $p_\mathrm{T}$ for different $R_\mathrm{T}$ classes in the toward (**top** panels) and transverse (**bottom** panels) regions, over the $R_\mathrm{T}$-integrated per-trigger yield.

In the c-jet triggered events a separation in relative yields is seen with $R_T$ in the toward region below the trigger threshold. However, unlike in the case of hadron-triggered events, the production of D mesons is more strongly dependent on $p_T$. The resemblance between these two trigger-type events shows that charm production in this region is generally disconnected from the leading hard process. The same observation can be made from the D-meson production in the transverse region in c-jet triggered events, where the qualitative behaviour is similar both in the jet-triggered and the hadron-triggered case. This also indicates that in general, charged-hadron triggers in events with a high-momentum charm select that charm as the leading hard process. There is, however, a striking difference in the behaviour of the curves above the trigger threshold: instead of all the curves being consistent with unity, lower underlying-event activity corresponds to a higher D-meson yield. (Note that a similar pattern is present for B-meson production in the case of b-jet triggered events). The probable cause of this phenomenon is the autocorrelation from wide-angle gluon-splitting processes. In cases when the gluon splits into a c$\bar{\text{c}}$ pair so that one falls into the toward and the other to the transverse region, the multiplicity will increase in the transverse region while the trigger will be fired, therefore the number of D mesons on average will be less. This effect will not show up prominently in the charged-hadron triggered case where a more energetic hadron is expected from the jet that balances the splitted gluon. However, further studies are required to verify this argument.

The right-hand side panels in Figure 4 show per-trigger D-meson production with light-jet triggers for the toward (top right) and transverse (bottom right) regions. We observe a separation in the yields from the different $R_T$ regions, meaning that these D mesons were produced in a secondary (heavy-flavour) hard process that is independent from the leading (light-flavour) hard process.

### 3.4. Effect of Charm and Beauty Fragmentation

Fragmentation causes a flavour-dependent shift in the momentum scale. To pin down the extent of this effect, in Figure 5 we compare production of charm and beauty at the partonic and hadronic levels, in terms of the underlying-event activity for charged-hadron triggered events. While the trends described for the D-mesons in Sections 3.2 and 3.3 are the same for both particle levels, in case of the charm quarks the boundary between the ranges dominated by the soft and hard production is at about 40% higher $p_T$ than in the hadronic case, rising from $p_T \approx 7 \, \text{GeV}/c$ for D-mesons to $p_T \approx 10 \, \text{GeV}/c$ for c-quarks in the charged-hadron trigger case. This is purely the effect of the change in momentum scale during the fragmentation of c quarks into D mesons.

In case of beauty, the transition between UE-dominated and leading-process dominated production is at $p_T \approx 10$ to $15 \, \text{GeV}/c$ in the case of both the B mesons and the b quarks. This is due to the much higher mass of b quarks corresponding to a larger dead-cone effect, that results in a harder fragmentation of quarks into the B meson than in the c quark into D meson case.

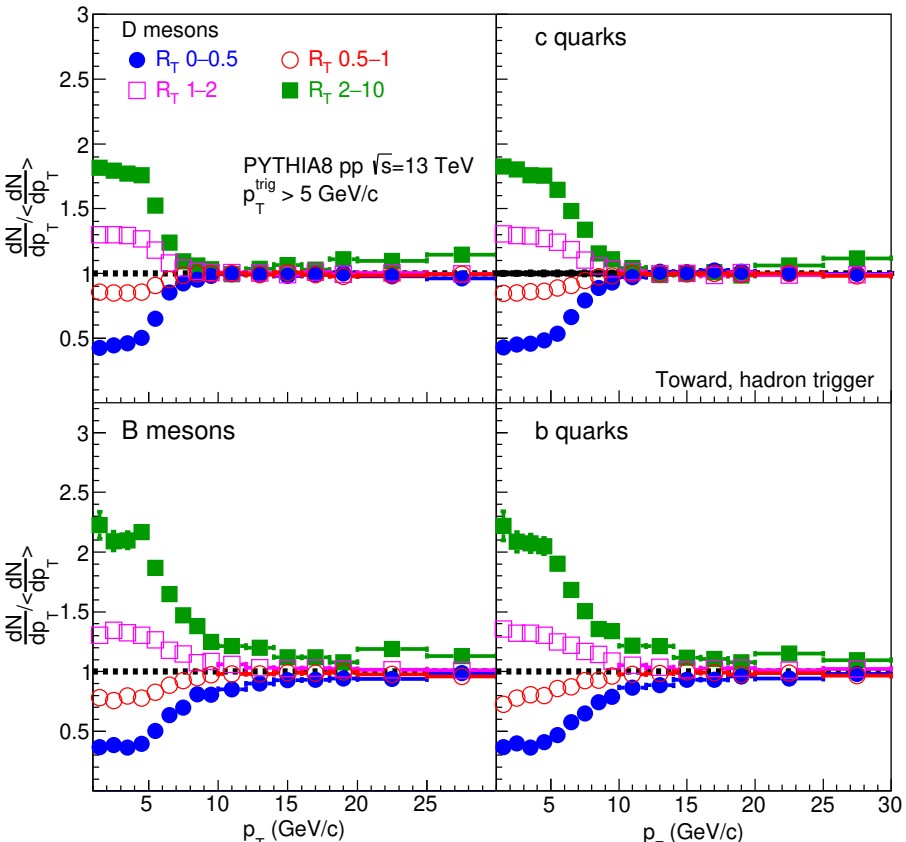

**Figure 5.** Per-trigger production ratios of D mesons (**top left**), c quarks (**top right**), B mesons (**bottom left**) and b quarks (**bottom right**) with charged-hadron triggers as a function of $p_T$ for different $R_T$ classes in the toward region, over the $R_T$-integrated per-trigger yield.

## 4. Conclusions

We presented a systematic analysis of heavy-flavour production in connection to a leading hard process in pp collisions at $\sqrt{s} = 13$ TeV, using transverse-event-activity differential simulations from PYTHIA 8. We conclude that the production of low-momentum heavy flavour in toward region in the events with hadron trigger is mostly determined by the underlying event.

However, while most heavy flavour in these events is produced in the initial hard process, heavy-flavour fragmentation and decay kinematics complicates the characterisation of the underlying event, and the usual method of using the leading charged final-state hadron as a trigger may wash away the connection between the leading process and the heavy-flavour hadron created in association with that.

In case of events triggered with light or heavy-flavour jets, however, this connection is retained and therefore they bring more direct information on the initial heavy-flavour production process. We showed that in charged-particle charm-jet triggers, the production of heavy flavour is $R_T$-dependent over the whole $p_T$ range, which is likely an effect of gluon radiation. Light-flavour jet triggers, on the other hand, provide means to trigger on a hard process that is different from that of heavy-flavour production, and therefore they allow for the underlying-event dependent analysis of the connection between two hard processes. We also investigated the impact of fragmentation by comparing the UE-differential production of c and b quarks to D and B mesons respectively. As expected, we see a $p_T$ shift caused by fragmentation, that is much smaller for the higher-mass b quarks than for the c quarks.

The upcoming Run-3 phase of LHC is opening up the possibility for high-precision and highly differential heavy-flavour measurements. The methods that we propose here

a provide great opportunity for the detailed verification of calculations for light and heavy-flavour production in the jet and the underlying event, and set the base for further model development.

**Author Contributions:** Conceptualization, R.V.; methodology, L.G. and S.S.; software, S.S. and L.G.; validation, R.V.; formal analysis, L.G. and S.S.; investigation, S.S. and L.G.; resources, R.V.; writing—original draft preparation, L.G. and S.S.; writing—review and editing, R.V.; visualization, L.G., S.S. and R.V.; supervision, R.V.; project administration, R.V.; funding acquisition, R.V. All authors have read and agreed to the published version of the manuscript.

**Funding:** This work has been supported by the NKFIH grants OTKA FK131979 and K135515, as well as by the 2019-2.1.11-TÉT-2019-00078 and 2019-2.1.6-NEMZ_KI-2019-00011 projects.

**Institutional Review Board Statement:** Not applicable.

**Informed Consent Statement:** Not applicable.

**Data Availability Statement:** Detailed results are available upon request from the Authors. Experimental data used in this study can be freely accessed at http://www.hepdata.net (accessed on 3 July 2022).

**Acknowledgments:** The authors acknowledge the computational resources provided by the Wigner GPU Laboratory and the research infrastructure provided by the Eötvös Loránd Research Network (ELKH).

**Conflicts of Interest:** The authors declare no conflict of interest.

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
