# Peer review of "Defining the Underlying-Event Activity in the Presence of Heavy-Flavour Processes in Proton-Proton Collisions at LHC Energies"

_2571-712X, doi:10.3390/particles5030021_

Round 1

Reviewer 1 Report

This review covers the article entitled, Defining the underlying-event activity in the presence of heavy-flavour processes is submitted by the authors.

Summary

-------

The authors present a systematic analysis of heavy-flavour production in the underlying event in connection to a leading hard process in pp collisions at s = 13 TeV, using the PYTHIA 8 with Monash tune Monte Carlo event generator. The authors proposed a method to verify in great details the model calculations for light and heavy-flavour production in the jet and the underlying event.

The authors simulated pp collisions at s = 13 TeV centre-of-mass energy using the Monte Carlo event generator PYTHIA 8 [19] with the Monash tune [20] and soft QCD settings. The complete analysis detail and various cuts used for analysis is presented in section 2 of the paper. The authors compare the results from events selected by triggering on the leading hadron, as well as those triggered with reconstructed jets. These results show that the kinematics of heavy-flavour fragmentation complicates the characterisation of the underlying event, and the usual method which uses the leading charged final-state hadron as a trigger may wash away the connection between the leading process and the heavy-flavour particle created in association with that. Events triggered with light or heavy-flavour jets, however, retain this connection and bring more direct information on the underlying heavy-flavour production process, but may also import unwanted sensitivity to gluon radiation.

The authors provide an extensive discussion on data set and analysis method. In summary, the authors point out that a new method proposed would be helpful to verify the details of model calculations for light and heavy-flavour production in jet and the underlying events.   

Review

------

This paper is well organized and clearly presented. The analysis is straightforward, and there is ample discussion of analysis techniques and contributing errors. The authors provide ample references of related work.

Of course it isn’t necessary that every or even most experimental papers include complete theoretical explanations, but it is obvious that the new results will be of interest to the theoretical community such that it will eventually lead to some new physics conclusion. Are there plans to perform these measurements from EPOS or any other best available model? At least one more plausible pathway to a physical interpretation needs to added to this manuscript before it can be accepted for publication or any reason why only one Pythia is helpful to study these measurements. Did the authors try to compare Pythia8 (default) with Pythia 8 (Monash) or Pythia (Perugia) tunes?

Minor Corrections

----------------------

Introduction Section Page-1

l-16 the so called ... à so called the Quark-gluon plasma

l-51 proton-proton à pp collisions

Analysis Method Page-3

l-97 Please add the spaces between the centrality intervals for example 0 – 5% and so on

l-107 proton-proton à pp collisions

Results Page 3

l-122 in function of à as a function of

l-156 and l-180 l-200 $c-\overline c$ à $c \overline c$

l-183 please recheck the sentence starting from “Note that …..” and rephrase as it is not clear

Reviewer 2 Report

COMMENTS TO THE AUTHOR(S)

Referee report on the Manuscript ID: particles-1776215

Entitled "Defining the underlying-event activity in the presence of heavy-flavor processes"

In this paper, the authors presented a very intuitive case of understanding heavy-flavor production in connection to underlying events, using a recently used observable in experiments called transverse event-activity classifier. Affects on heavy-flavor production due to the choice of events triggered by hadrons and jet is also presented. This study is made using PYTHIA 8 Monte Carlo event generator.

The paper is interesting and well written. Study of the effect of charm and beauty fragmentation is also worked out, considering production at partonic and hadronic levels is relatively fresh. However, some clarifications/improvements are needed. I would recommend the publication of the manuscript in the Journal Particle if the authors could clarify or make amendments wherever necessary, as detailed in the following points.

(1). Page-1 line 10. The word "proposed" may not be correct as this observable is already proposed in Eur. Phys. J. C, 76, 299 (2016)

(2). Page 2, line 73: "soft QCD setting", Since this analysis involve the production of heavy-flavored particles (D/B mesons), I wonder why not HardQCD tune of PYTHIA8 is used, followed by pT divergence cut and minimum pT value, instead of softQCD tuning.

(3). Page 2, line 80: What is the motivation for choosing trigger threshold pT to be above 5 GeV/c, is it inspired by experimental studies. A few sentences of explanation for using this value would be of particular interest to the general readers.

(4). Page 2, lines 83 and 85: Please provide a reference for choosing R =0.4 and threshold pT for trigger jet to be 10 GeV/c or motivation for this selection.

(5). There is no mention of the tuning parameters used in Pythia for the current study. As this work deals with underlying effects and PYTHIA8 has very much control over the different underlying effects, their choice will affect the final results. Authors should mention the generic details on the tuning (like MPI, mode of CR) used for the simulation to make a proper physics comment, like how the nature of initial/final-state quantities affects the results obtained. Also, this will give a scope to reproduce the results.

(5). Page 3, line 95: It would be interesting to see how well R_T distribution obtained in the experiment as given in JHEP04(2020)192 for pp@13 TeV, is explained by the present study.

(6). What is the value of <N_trans> used here for the tuning of Pythia? The literature shows that <N_trans> is sensitive to the different Pythia tuning. And how close/far is this value from literature like Eur. Phys. J. C, 76, 299 (2016).

(7). Page 3, line 122-124:  Authors have compared their simulation with ALICE result, which is done for light-flavor production, and the current study is done for heavy-flavor. In general, for charmed/bottom-based hadrons in Pythia8, HardQCD is evoked as they involve initial HardQCD processes like gluon splitting, etc. I wonder how the Pythia setting used here for comparing results with ALICE is different from what is used further in this work for heavy-flavored study?

(8). Page 4, line 149: "hadron triggers ...", by hadron, do authors mean all the final-state charged hadrons (pion, kaons, and protons) or is D-meson with pT > 5 Gev/C is also counted as hadron trigger?

(9). Page 4, line 169-171: The experiment mentioned in Ref. 27 of the current manuscript has an uppercut to pT, i.e., pT < 40 GeV/c. Is such an upper cut considered in this present work?

(10). A minor suggestion: I would modify paper title to give information about system, energy and model used for this analysis.

*******************************************************************************

Reviewer 3 Report

The presented research results clearly show an importance of the UE in jet studies (especially for heavy flavour jets). The paper in its current form reads nicely but, in my opinion, can be improved. One can find a list of comments/questions and suggestions below.

1) In the introduction authors mention that possible collectivity in small systems will play a substantial role and modify both UE and jets. Nevertheless, in this paper authors test only the default Monash tune of PYTHIA. Already in reference [14] one can find that spectra at high pT depends significantly on possible PYTHIA tunes. It would be also important to test how colour rope formation and string shoving would modify the results. I suggest to add a comparison of default Monash tune results with the results obtained with string shoving and rope formation with the latter being especially influential on heavy flavours.

2) In ALICE paper [13] there are PYTHIA calculations similar to Fig1. Is there any difference with predictions obtained in this paper? Isn't it a repetition? If so - is it needed in this paper?

3) It would be helpful for readers to have a look at RT distributions in order to understand why the certain RT classes were selected. 

4) In [CMS, Eur.Phys.J.C 81 (2021) 4, 312] it is suggested to split the transverse region in transverse min and transverse max by looking at the least and most charged-particle activity. It might be helpful to perform the same separation in your studies. 

Round 2

Reviewer 2 Report

The authors have answered all of my comments in detail and improved
the manuscript substantially. I recommend this manuscript for the publication in the Journal Particles.

Reviewer 3 Report

I would like to thank authors for their detailed replies. In particular, I find the observation stated in the point 4of the  reply very interesting. In my opinion, it is indeed worth to investigate it in more details.